# The transcription-repair coupling factor Mfd associates with RNA polymerase in the absence of exogenous damage

Han N. Ho [1], Antoine M. van Oijen [1] & Harshad Ghodke [1]

During transcription elongation, bacterial RNA polymerase (RNAP) can pause, backtrack or stall when transcribing template DNA. Stalled transcription elongation complexes at sites of bulky lesions can be rescued by the transcription terminator Mfd. The molecular mechanisms of Mfd recruitment to transcription complexes in vivo remain to be elucidated, however. Using single-molecule live-cell imaging, we show that Mfd associates with elongation transcription complexes even in the absence of exogenous genotoxic stresses. This interaction requires an intact RNA polymerase-interacting domain of Mfd. In the presence of drugs that stall RNAP, we find that Mfd associates pervasively with RNAP. The residence time of Mfd foci reduces from 30 to 18 s in the presence of endogenous UvrA, suggesting that UvrA promotes the resolution of Mfd-RNAP complexes on DNA. Our results reveal that RNAP is frequently rescued by Mfd during normal growth and highlight a ubiquitous house-keeping role for Mfd in regulating transcription elongation.

---

[1] School of Chemistry, University of Wollongong, and Illawarra Health and Medical Research Institute, Wollongong, NSW 2522, Australia. Correspondence and requests for materials should be addressed to H.G. (email: harshad@uow.edu.au)

Transcription is a complex metabolic process in which genetic information encoded in double-stranded DNA (dsDNA) is transcribed by the RNA polymerase (RNAP) to form RNA. In bacteria, this reaction is executed by the bacterial RNAP composed of five subunits $\alpha_2\beta\beta'\omega$[1]. Whereas these core proteins catalyse the templated addition of rNTPs to the growing RNA chain, additional accessory factors are required to initiate and execute successful rounds of transcription in cells. The activities of these factors are coordinated to cycle RNAP through the various stages of the transcription cycle, a process that includes promoter recognition, closed complex formation, open complex formation, abortive initiation, promoter escape, elongation and termination[2].

Far from being a highly processive reaction, transcription elongation is punctuated by nucleotide mis-incorporations, pausing or stalling events and collisions with template-bound proteins acting as roadblocks[3–6]. These events may lead to transcription elongation complexes (TECs) entering non-productive states that leave it stalled on the DNA. As highly stable complexes, failed TECs can pose a threat to cell survival. While in some cases RNAP may be able to resume elongation unaided, in *Escherichia coli* (*E. coli*) these stalled TECs are often rescued by the concerted action of several transcription modulators such as GreA/B[7], NusA and UvrD[8], and Mfd[9]. Depending on particular transcription modulator(s) associated with the stalled RNAP, RNAPs are led into various pathways including transcript cleavage and restart, or transcription termination by forward or backward translocation. Understanding the interplay between various regulatory factors that enable processive and high-fidelity transcription has been an area of intense investigation and continues to be a long-standing challenge in the context of live cells[10–12].

In *E. coli* the transcription-repair coupling factor Mfd promotes transcription termination at sites of stalled RNAP and recruits the nucleotide excision repair (NER) machinery to the site of the damage (reviewed in ref. [13]). This transcription-coupled repair (TCR) pathway occurs in the following manner: first, Mfd binds the β subunit of stalled RNAP at the upstream side of the transcription bubble[14,15]. Second, RNAP binding is thought to promote a conformational change in Mfd that allows it to bind dsDNA and trigger its translocase activities[16]. Third, translocation along dsDNA results in the collapse of the transcription bubble and enables dissociation of RNAP from the template DNA, and the nascent transcript from RNAP[17]. In the final step, the lesion is handed over from RNAP to the UvrAB proteins that specifically recognize the DNA damage and coordinate repair[9].

Whereas the mechanism of TCR is reasonably well understood from several in vitro studies, it is not clear how the different partners interact in the intracellular environment, where multiple DNA-repair factors often perform redundant functions. With the objective of localizing and measuring the kinetics of association of Mfd with RNAP in live cells, we constructed an MG1655 strain carrying a chromosomal fusion of the bright fluorescent protein YPet to the C-terminus of Mfd. We found that fluorescently tagged Mfd retains the ability to complement transcription-coupled NER in live cells when assayed for UV survival. Live-cell imaging of growing bacteria revealed a pervasive association of Mfd with the nucleoid even in the absence of exogenously induced DNA damage. Using a combination of small-molecule modulators of transcription and mutants of Mfd, we demonstrate that this extensive association of Mfd with the nucleoid arises primarily from interactions of Mfd with stalled TECs of RNAP. These observations demonstrate that Mfd rescues stalled RNAPs during normal metabolism, in the absence of externally inflicted DNA damage. Based on our observations, we conclude that Mfd serves a critical house-keeping role in regulating transcription during normal growth.

## Results

**Visualization of Mfd in live *E. coli* cells.** To visualize Mfd in live *E. coli*, we employed λ Red recombination to create a C-terminal chromosomal fusion of *mfd* with the gene for the bright yellow fluorescent protein YPet[18]. This strain expresses the Mfd-YPet fusion protein from its native promoter (Fig. 1a). To identify whether the fusion protein retains the ability to facilitate the TCR of UV-induced damage in live cells, we performed UV-survival assays. Specifically, we compared growth rates of Δ*recA* cells carrying *mfd*, *mfd-ypet* or Δ*mfd* in the MG1655 background after exposure to 254-nm light (Fig. 1b). Ultraviolet light is a strong inducer of the SOS DNA-damage response in bacterial cells. Upon SOS, the expression of the UvrABD and Cho—DNA-repair factors that execute the global genomic nucleotide-excision repair pathway[19]—is upregulated[20]. The increase in copy number of UvrA by approximately tenfold prioritizes the detection of UV lesions via the global genomic NER pathway, consequently masking the signal from TCR[21]. Therefore, we assayed for Mfd-YPet function in UV-sensitive Δ*recA* cells that do not exhibit SOS induction in response to DNA damage. In this genetic background, *mfd*⁻ cells have been demonstrated to be significantly more UV-sensitive than *mfd*⁺ cells[9]. Following UV exposure, *mfd-ypet* Δ*recA* cells recovered at similar rates to *mfd*⁺ Δ*recA* at low doses (0 and 1 J m⁻²) while slight delays were observed at 2.5 and 5 J m⁻² (Fig. 1b). On the other hand, Δ*mfd* Δ*recA* cells exhibited delay in recovery at intermediate UV doses (1 and 2.5 J m⁻²) and failed to recover upon exposure to 5 J m⁻². These results demonstrate that the C-terminal YPet fusion retains function and is able to complement the repair of UV-induced lesions.

Next, we performed live-cell imaging to visualize the intracellular localization of Mfd. Cells expressing Mfd-YPet from the native chromosomal locus (*mfd-ypet*) were grown in EZ-rich defined medium and deposited on a silanized coverslip in a flow cell. This setup enabled us to visualize fast growing cells in the presence of fresh growth medium at 30 °C (Fig. 1c, Methods and Supplementary Fig. 1). Illumination with 514-nm laser light revealed high fluorescence signal in cells that was attributable to a high concentration of Mfd-YPet. Relying on the single-molecule sensitivity of our fluorescence imaging, we determined the copy number of fluorescent Mfd-YPet in our strain using photobleaching experiments. By measuring the initial fluorescence intensity of the cells and dividing it by the average intensity of a single YPet molecule, we were able to estimate the copy number of Mfd-YPet to be 22 ± 5 per cell (mean ± s.d., 254 cells; see Supplementary Fig. 2a−c and Supplementary Note 3). While the copy number of Mfd is widely believed to be 500 copies per cell[22], mass spectrometry-based measurements have determined Mfd abundance in *E. coli* MG1655 grown in LB medium to be 48 ± 5 copies per cell (mean ± s.d.)[23]. Our estimate is consistent with the latter measurement and likely reflects an underestimate of the true copy number of Mfd for two reasons. First, slow maturation or mis-folding of the fluorophore will result in dark Mfd-YPet molecules that are invisible in these assays[24]. Second, C-terminally tagged Mfd may be expressed with lower efficiency compared to wild-type Mfd. The slight delay in recovery of *mfd-ypet* cells compared to *mfd*⁺ following exposure to UV doses of 2.5 and 5 J m⁻² (Fig. 1b) may be attributable to a potentially lower expression level of Mfd-YPet compared to that of untagged Mfd in wild-type cells.

Consistent with its role as a DNA-repair factor, Mfd-YPet displayed a nucleoid-associated signal[25] (Fig. 1d and

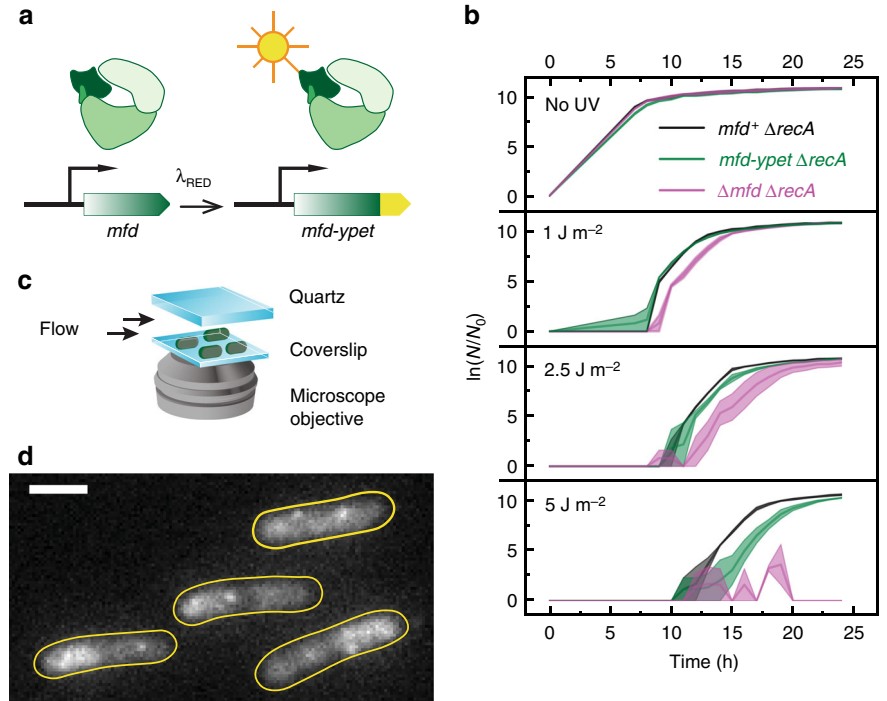

**Fig. 1** Visualization of Mfd in live *Escherichia coli*. **a** Schematic of construction of *mfd-ypet* strain using λ Red recombination. The native gene is replaced by a chromosomal *mfd-ypet* fusion under its native promoter. **b** The chromosomal *mfd-ypet* fusion allele retains function when assayed for UV-survival. Growth curves of *mfd*⁺ Δ*recA* (black), *mfd-ypet* Δ*recA* (green) and Δ*mfd* Δ*recA* (magenta) following exposure to various doses of 254-nm light. $N_O$ and $N$ are the numbers of cells at the beginning of the experiment and at the time of measurement respectively. Shaded error bars represent standard error of the mean of two measurements, with each measurement being an average of two technical replicates. **c** Schematic of experimental setup for visualizing Mfd-YPet in live cells. **d** Representative fluorescence image of *mfd-ypet* cells upon illumination with 514-nm light. Image is an average projection of ten continuous 100-ms frames. Cell outlines (yellow) were drawn based on the bright-field image. Scale bar, 2 μm

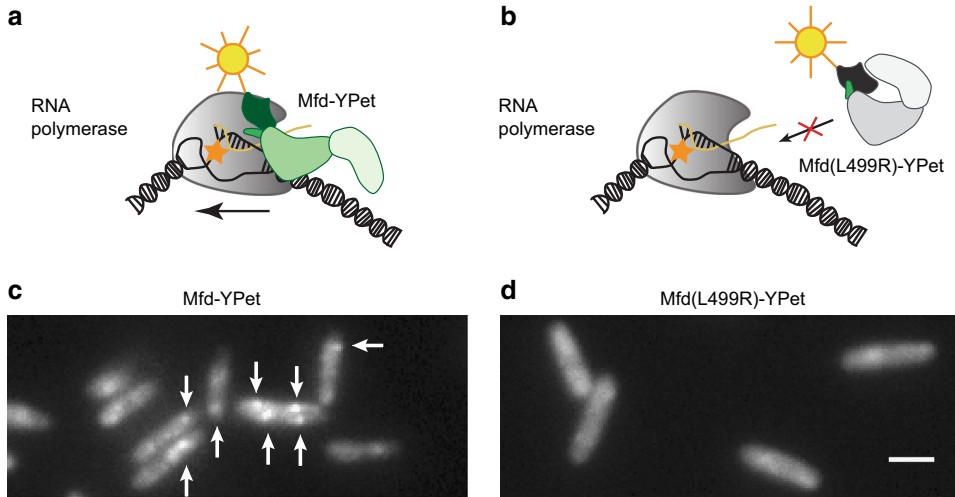

**Fig. 2** Mfd-YPet associates with the nucleoid via interactions with RNA polymerase. Schematic describing the initial stages of transcription-coupled repair. **a** Mfd is recruited to RNA polymerase stalled at bulky lesions (orange star) on the transcribed strand. The arrow indicates the direction of transcription elongation by RNA polymerase. **b** Substitution of L499 residue with arginine (R) results in the mutant Mfd(L499R) that is unable to rescue stalled transcription complexes. Fluorescence images of Δ*mfd* cells expressing plasmid-based **c** wild-type Mfd-YPet or **d** mutant Mfd(L499R)-YPet. White arrows highlight well-defined Mfd-YPet foci. Each image is an average projection of ten continuous 100-ms frames. Scale bar, 2 μm

Supplementary Movie 1). The Mfd-YPet distribution was found to be heterogeneous, with regions of high intensity, reminiscent of the localizations of RNA polymerase in live bacterial cells[26]. Additionally, these data revealed that Mfd-YPet forms foci in these cells, suggesting that the localizations visualized in our experiments arise out of multiple transient or stably bound Mfd molecules interacting with the nucleoid.

**Binding of Mfd is mediated via interactions with RNAP**. We next probed whether the Mfd-YPet foci visualized in our

experiments represented associations of Mfd with RNAP on DNA (Fig. 2a). This association could be demonstrated by visualizing an Mfd mutant that is impaired in its interactions with RNAP and unable to displace stalled RNAP. We selected the Mfd (L499R) mutant as it has previously been demonstrated to be deficient in interactions with the β-subunit of RNAP, while retaining other functions such as DNA binding, nucleotide binding and ATP hydrolysis[14] (Fig. 2a, b). To that end, we constructed a low-copy plasmid expressing Mfd(L499R)-YPet under the native *mfd* promoter (Supplementary Note 2) and imaged it in Δ*mfd* cells. In control studies, we expressed wild-type Mfd-YPet from the same low-copy plasmid, under its native promoter (Supplementary Note 2). In these cells, the copy numbers of Mfd-YPet and Mfd(L499R)-YPet were found to be 190 ± 40 (mean ± s.d., 232 cells) and 170 ± 40 (mean ± s.d., 183 cells) respectively (Supplementary Fig. 2d, e).

Live-cell imaging of cells expressing wild-type Mfd-YPet from the plasmid reveals a similar localization pattern as Mfd-YPet expressed from the chromosome, with defined foci in most cells (Fig. 2c). On the other hand, Mfd(L499R)-YPet exhibited a homogenous cytosolic distribution (Fig. 2c, d and Supplementary Fig. 3a−c). These results suggested that Mfd(L499R)-YPet does not bind stably to its substrates in live cells. Upon closer examination, Mfd(L499R)-YPet was found to exhibit foci on the 100-ms timescale reflective of transient interactions (Supplementary Movie 2). The loss of defined foci upon the introduction of the disruptive L499R mutation in the surface of the RNAP-interacting domain suggests that the stable binding events visualized in the experiments for wild-type Mfd arise primarily from interactions with RNAP. Further, these results also demonstrate that RNAP is the major substrate of Mfd in live cells under conditions of normal growth.

**Mfd primarily associates stably with elongating complexes**. To further probe the association of Mfd with RNAP, we used small-molecule modulators of transcription that are known to interact with specific conformations adopted by RNAP during the transcription cycle. The anti-microbial drug rifampicin (Rif) has been demonstrated to selectively target the bacterial transcription initiation complex[27]. Binding of Rif to RNAP sterically blocks the path of the nascent RNA and prevents transcription elongation beyond three nucleotides[27]. By doing so, Rif selectively inhibits the transition to TECs and does not influence the ability of RNAP to bind promoters and form open complexes[28,29]. Since actively elongating RNAP complexes are refractory to Rif, we treated cells with Rif for 60 min, to ensure that any ongoing rounds of RNA synthesis are terminated by 60 min and all TECs have been removed. Visualization of Mfd-YPet in Rif-treated cells revealed a homogenous distribution of the Mfd-YPet signal (Fig. 3b, c and Supplementary Fig. 4).

To quantify binding events of single Mfd-YPet molecules, we collected rapid-acquisition movies of Mfd-YPet cells. In this format, we make use of the fact that single YPet molecules stochastically return to the bright state after photobleaching[26,30]. The imaging protocol can be divided into two phases: Phase I corresponding to the decay in fluorescent signal caused by photobleaching and Phase II corresponding to the reactivation phase. In the first phase of imaging, the cellular YPet fluorescence disappeared in the first 30 frames (equivalent to 3 s using 100-ms frames) (Fig. 3a and Supplementary Movie 1). In Phase II (frames 31−50), single YPet molecules were observed to return to the bright state (Fig. 3e, left panel, and Supplementary Movie 1). The reduced cytosolic background enabled single Mfd-YPet molecules to be imaged and tracked unambiguously. To quantify Mfd-YPet binding upon treatment with Rif, we counted Mfd-YPet foci per

cell that could be tracked in consecutive frames for at least 200 ms in Phase II (referred to as 'sub-second foci'). The number of Mfd-YPet sub-second foci reduced sevenfold (Fig. 3e, f) in Rif-treated cells compared to untreated cells. These observations demonstrate that the genome-wide, stable association of Mfd with the nucleoid is largely lost when RNAP is unable to form elongating complexes.

Interpreting the fluorescence foci as RNAP-associated Mfd leads to the prediction that drugs promoting the stalling of TECs should promote stable association of Mfd-YPet and thus increase the lifetime of the foci. CBR703 is a prototypical member of a class of RNAP inhibitors that inhibit nucleotide addition during the catalytic step of transcription[31]. In this activity, it binds the β′ bridge helix interconnecting the two largest RNAP subunits and promotes RNAP pausing[32]. Since Mfd is involved in recognizing stalled or paused RNAPs, we expected that cells treated with CBR703 would exhibit an increase in the number of Mfd-YPet foci. Since the potency of CBR703 is greater in *tolC* mutants of *E. coli*[33], we visualized Mfd-YPet expressed from its chromosomal locus in Δ*tolC* cells upon exposure to 75 µg per mL of CBR703 after 30 min of incubation in a flow cell. As expected, averages of frames 1–10 (corresponding to a total of 1 s exposure time) in Phase I of imaging revealed a greater number of Mfd-YPet foci (Fig. 3d) as compared to untreated cells (Supplementary Fig. 5a −c). Examination of foci detected in Phase II of imaging also revealed a greater number of Mfd-YPet sub-second foci per cell compared to untreated wild-type cells or Rif-treated cells (Fig. 3e, f). Control experiments with Mfd(L499R)-YPet cells treated with CBR703 confirmed that the greater number of CBR703-dependent localizations do not arise out of an off-target effect of drug treatment (Supplementary Fig. 5d). Additionally, we detected a greater number of immobile RNA polymerase foci in an MG1655 *tolC* mutant strain expressing fluorescently labelled RNA polymerase following CBR703 treatment (Supplementary Fig. 6). Collectively, our experiments confirm that Mfd associates with TECs in live cells.

These observations leave two possibilities that describe interactions of Mfd with TECs: either Mfd associates with RNAP constitutively during transcription or Mfd associates with TECs that are stalled. Wild-type Mfd-YPet cells exhibited a localization that was a mixture of well-defined foci and diffuse cytosolic signal suggesting that only a fraction of the population of Mfd is bound to nucleoid-associated RNAP at any time, and that unbound Mfd is diffusive in the cytosol (Fig. 3b). Upon treatment with Rif, the Mfd signal associated with the nucleoid via interactions with RNAP was lost due to the inability of RNAP to form TECs. Rif-treated cells exhibit a largely homogenous Mfd signal distribution (Fig. 3c). In the presence of CBR703 (Fig. 3d), we see this equilibrium driven to the opposite extreme—a greater number of Mfd-YPet foci are detected upon drug treatment, attributable to a greater number of drug-stalled RNAPs. These results reveal that under conditions of fast growth with several hundreds to thousands of RNAP engaged with the nucleoid in transcription elongation[34,35], only a fraction of Mfd is associated with TECs. Considering previous observations that Mfd plays a role in rescuing stalled transcription complexes[9] and the observation that stable Mfd association with the nucleoid is enriched in the presence of CBR703, we interpret the Mfd visualized in our experiments as associated with stalled RNAPs.

**Interactions of Mfd with TECs are long lived**. DNA-repair factors often possess multiple binding modes that enable specific and non-specific interactions with their substrates[36–38]. Therefore, we hypothesized that the Mfd-YPet population detected in cells reflects sub-complexes of Mfd associated with RNAP during

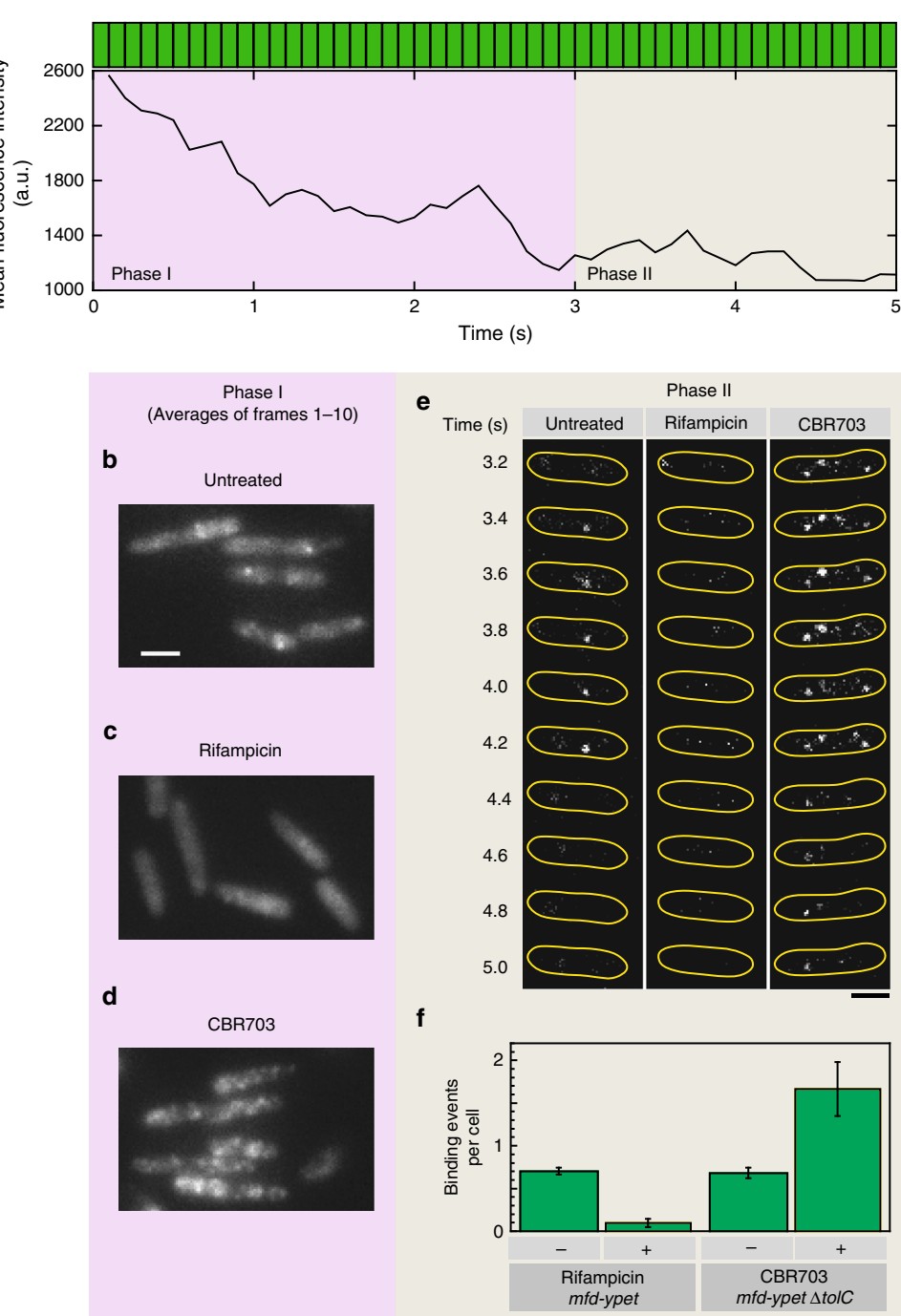

**Fig. 3** Transcription inhibitors change Mfd-YPet binding frequency. **a** Mean cellular fluorescence intensity of a representative single *mfd-ypet* cell during a 5-s continuous acquisition in 514-nm channel (each green bar (top) represents a 100-ms frame). To visualize single molecules of Mfd-YPet, the cellular fluorescence was first photobleached in a bleaching phase (Phase I; 3 s) followed by a photo-reactivation phase (Phase II; 2 s). Average projections of the first ten frames in the bleaching phase (Phase I) of *mfd-ypet* cells **b** untreated or **c** rifampicin-treated (50 μg per mL, 60 min) and **d** *mfd-ypet ΔtolC* cells with CBR703 treated (75 μg per mL, 30 min). **e** Montage of 100-ms frames during photo-reactivation phase from an *mfd-ypet* cell (untreated) or treated with rifampicin, and an *mfd-ypet ΔtolC* cell treated with CBR703. **f** Bar plot representing the number of Mfd-YPet sub-second binding events detected in untreated *mfd-ypet* cells (n = 1564 cells), cells treated with rifampicin (n = 396 cells), *mfd-ypet ΔtolC* untreated (n = 966 cells) and in CBR703 (n = 598 cells). Error bars are standard deviations from three independent experiments for each condition. Scale bars, 2 μm

various stages of TCR. We set out to measure the residence times of various sub-complexes of Mfd that occur in live cells. Tracking single molecules of fusion proteins tagged with genetically expressible fluorophores on long time scales in live cells is challenging due to the poor photo-stability of fluorescent proteins. To extend observation of bound molecules to the time scale of several minutes, we employed an interval-based imaging strategy that involves addition of dark frames between

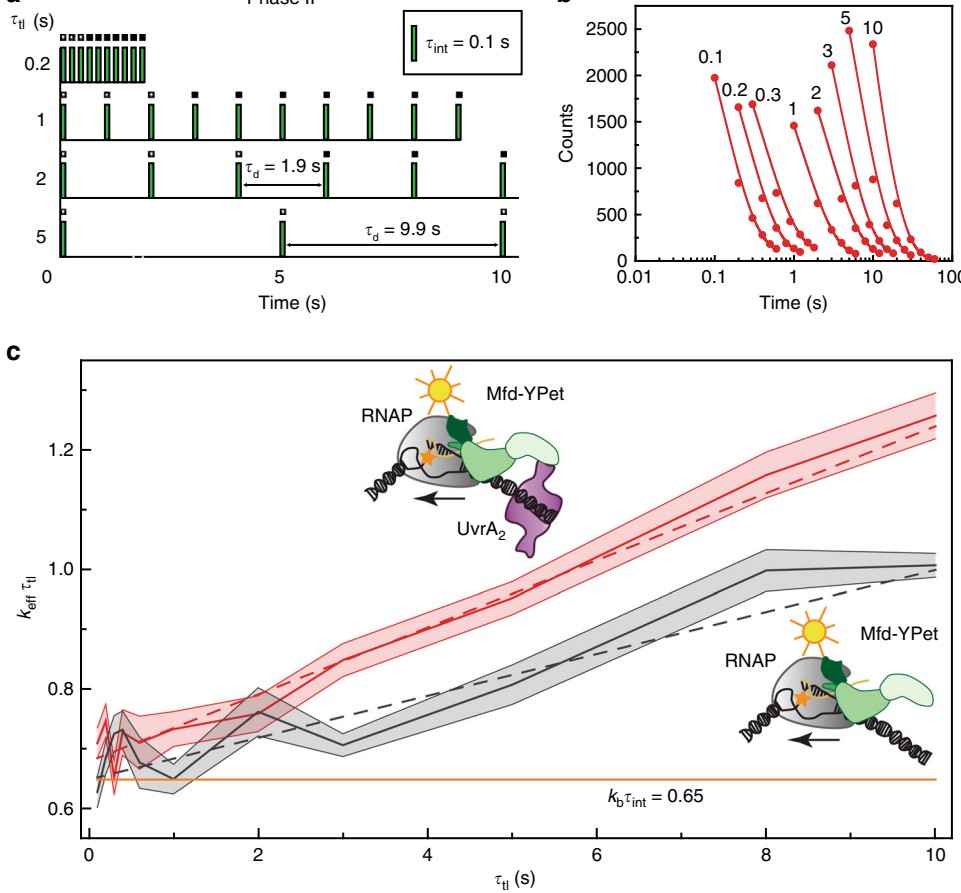

**Fig. 4** Kinetics of association of Mfd-YPet with RNA polymerase in live cells. **a** Intracellular off-rate constants of Mfd-YPet were measured using an interval imaging scheme. Imaging was performed in two phases: in the first phase, continuous illumination was applied to bleach the signal followed by a reactivation phase (shown here) where photo-reactivated molecules were visualized. In this imaging scheme, a dark frame with a varying duration ($\tau_d$) was inserted between consecutive acquisition frames (green bars, $\tau_{int} = 0.1$ s) in phase II (reactivation phase). Time-lapse time ($\tau_{tl}$) is the sum of $\tau_{int}$ and $\tau_d$. **b** Cumulative residence time distributions of Mfd-YPet foci in *mfd-ypet uvrA+* for various time-lapse times ($\tau_{tl}$; values are indicated above each trace). Red circles represent combined observations from nine independent experiments. Effective off-rate constant ($k_{eff}$) is obtained from the exponential fit (line) to the experimental data at each time-lapse time. **c** $k_{eff}\tau_{tl}$ vs. $\tau_{tl}$ plot for the determination of the off-rate constants ($k_{off}$) of Mfd-YPet in *mfd-ypet uvrA+* (red curve) and *mfd-ypet ΔuvrA* (grey curve) cells. Shaded error bars represent standard deviations of the bootstrap distribution of $k_{eff}\tau_{tl}$ for the corresponding $\tau_{tl}$. Dash lines represent linear least squares fits to the corresponding data. The orange line (continuous) represents a hypothetical static binding with $k_{off}$ equal to zero and $k_b\tau_{int}$ equal to 0.65. Cartoons (insets) represent possible Mfd-RNAP complexes forming in *mfd-ypet uvrA+* (top) and *mfd-ypet ΔuvrA* (bottom) cells

acquisitions. This strategy has been recently used to interrogate binding lifetimes of transcription factors and other DNA-interacting proteins in live cells[39–41] (Methods and Supplementary Note 4).

We employed the two-phase imaging strategy described before with one important difference: in Phase II of imaging, we added an additional dark period ($\tau_d$) between two consecutive frames (with integration time ($\tau_{int}$) of 0.1 s) (Fig. 4a). In practice, we collected several acquisitions of growing cells under identical illumination conditions while changing the time-lapse time (defined as $\tau_{time\text{-}lapse}$ ($\tau_{tl}$) $= \tau_d + \tau_{int}$) in the range from 0.1 to 10 s. The low cellular fluorescence in Phase II enabled us to visualize and track bound, photo-reactivated Mfd-YPet molecules unambiguously.

The residence times of individual foci detected in each acquisition were measured and a cumulative residence time distribution was generated (Fig. 4b, Methods and Supplementary Table 4). This distribution describes the probability of a loss of a fluorescent focus as a function of the time interval (in real time).

In our live-cell measurements, this cumulative residence time distribution reflects the cumulative probability of two distinct phenomena: the probability of dissociation of Mfd from the RNAP-DNA complex or Mfd-RNAP complex from DNA, and probability of loss of the focus due to photobleaching of the fluorophore. Whereas the off-rate constant ($k_{off}$) is characteristic of the Mfd−RNAP interaction, the photobleaching rate constant ($k_b$) depends on the excitation intensity. Across data sets where only the time-lapse time is varied, the photobleaching rate (in frame time) is identical for all imaging conditions and can be de-convoluted from the off rates as demonstrated previously[39].

The cumulative residence time distributions were then fitted to a single-exponential model to obtain effective off-rate constants ($k_{eff}$) that represent mixtures of the photo-bleaching rate constant ($k_b$) and the off-rate constant ($k_{off}$) (Methods and Supplementary Note 4). In case of a two-state system, $k_{eff}\tau_{tl}$ and $\tau_{tl}$ follows a linear relationship where the slope reveals $k_{off}$ for the state-transition and the intercept provides $k_b\tau_{int}$ (Methods, Supplementary Note 4 and Supplementary Fig. 7)[39]. Linear least squares

**Table 1 Sub-second and transient foci per cell exhibited by Mfd-YPet and mutants in untreated and Rif-treated cells**

| Strain | Mfd and mutants | Copy number | Sub-second foci per cell | | | Transient foci per cell | | |
|--------|-----------------|-------------|-----------|-------------|----------------------------------|-----------|-------------|----------------------------------|
| | | | Untreated | Rif-treated | Fraction of Rif-insensitive foci (%) | Untreated | Rif-treated | Fraction of Rif-insensitive foci (%) |
| *mfd-ypet* | Mfd-YPet | 22±5 | 0.70±0.04 (n=1564) | 0.10±0.05 (n=396) | 14 | — | — | — |
| Δ*mfd* | Mfd-YPet | 190±40 | 1.4±0.7 (n=313) | 0.5±0.2 (n=430) | 36 | 1.0±0.4 (n=319) | 0.4±0.1 (n=544) | 40 |
| Δ*mfd* | Mfd(E730Q)-YPet | 140±30 | 2.6±0.4 (n=329) | 2.0±0.9 (n=291) | 77 | 2.4±0.5 (n=409) | 2.3±1 (n=285) | 96 |
| Δ*mfd* | Mfd(L499R)-YPet | 170±40 | 0.2±0.1 (n=371) | 0.19±0.04 (n=463) | 95 | 0.11±0.02 (n=407) | 0.09±0.05 (n=468) | 82 |
| Δ*mfd* Δ*uvrA* | Mfd-YPet | 190±40 | 1.0±0.2 (n=223) | 0.4±0.1 (n=317) | 40 | 0.9±0.5 (n=242) | 0.4±0.1 (n=390) | 44 |

Fraction of Rif-insensitive foci is defined as the ratio of foci per cell in Rif-treated and untreated cells. Mean ± s.d. (three experiments). n, the number of cells.

minimization of the Mfd-YPet data to a single off-rate model revealed a $k_{off}$ equal to $0.056 \pm 0.003\ \mathrm{s}^{-1}$, which corresponds to a residence time of about 18 s (Fig. 4c and Supplementary Table 6).

Does this measured lifetime reflect an on-pathway intermediate in TCR? Since UvrA is downstream of Mfd in the transcription-coupled NER pathway, and has been shown to promote the disassembly of Mfd-RNAP complexes[42], we asked if the residence time of Mfd in live cells is influenced by UvrA. We imaged *mfd-ypet* Δ*uvrA* cells using the interval imaging approach described above to measure the residence time of Mfd-YPet in the absence of UvrA. Kinetic analysis of Mfd-YPet binding in Δ*uvrA* cells revealed a single slowly dissociating species with a $k_{off}$ equal to $0.035 \pm 0.004\ \mathrm{s}^{-1}$ (Fig. 4c and Supplementary Tables 5,6), or a lifetime of nearly 30 s. In the presence of UvrA, the lifetime of Mfd-YPet was almost twofold faster (18 s for *uvrA*+ cells), demonstrating that UvrA (or a complex containing UvrA) promotes dissociation of Mfd in cells. Taken together, these experiments reveal that the 18 s lifetime of Mfd-YPet is governed by interactions with RNAP as well as UvrA, signifying that the repair intermediate described here corresponds to the lesion-handover complex in transcription-coupled NER.

**Mfd also associates at sites other than stalled TECs.** One intriguing observation from our experiments testing the association of Mfd with RNAP is that approximately 14% of Mfd-YPet foci (0.1 out of 0.7 binding events per cell, see Fig. 3f) remain in Rif-treated cells. To better understand these interactions, we imaged Mfd-YPet expressed from a low-copy plasmid in Δ*mfd* cells and quantified the number of Rif-sensitive and Rif-insensitive foci observed in reactivation phase (Phase II, see Methods). We counted the number of foci per cell that lasted for either 200 ms (termed 'sub-second') or 1 s in real-time (two 100 ms exposures with a 900 ms dark interval, henceforth referred to as 'transient'). As described above, the Mfd-YPet concentration in these cells is ninefold higher than that in *mfd-ypet* cells. In these cells, Rif-treatment caused a 64% reduction in sub-second binding events per cell from $1.4 \pm 0.7$ to $0.5 \pm 0.2$ (Table 1 and Fig. 5a). The twofold increase in the fraction of Rif-insensitive foci (compare 14% to 36%) in response to a ninefold increase in Mfd-YPet concentration indicates binding to a secondary substrate that is dependent on the expression level of Mfd-YPet.

Could the Rif-insensitive foci visualized in these experiments be attributable to dsDNA binding independent of RNAP? Structural and small angle X-ray scattering studies have revealed that Mfd normally adopts the auto-inhibited 'closed' state that exists in a dynamic equilibrium with the 'open' state[43].

Nucleotide binding promotes the formation of the open state that exhibits robust dsDNA binding[17,43,44]. Additionally, translocation activity is stimulated by binding to stalled TECs[16]. Therefore, to characterize DNA binding properties of Mfd in live cells, we attempted to tip the conformational equilibrium to the open state by introducing the E730Q mutation in the Walker B motif[43]. Mfd(E730Q) has been demonstrated to be deficient in ATP hydrolysis but not ATP binding, and exhibits similar dsDNA binding properties as Mfd bound to ATPγS in the absence of RNAP [43]. When expressed from a low-copy plasmid (140 ± 30 copies of Mfd(E730Q)-YPet per cell, see Supplementary Fig. 2f and Supplementary Fig. 3d), the E730Q mutant exhibited almost twofold more foci per cell than Mfd-YPet (sub-second foci: $2.6 \pm 0.4$ vs. $1.4 \pm 0.7$; transient foci: $2.4 \pm 0.5$ vs. $1.0 \pm 0.7$, see Table 1 and Fig. 5b). Following Rif-treatment, the frequency of sub-second Mfd(E730Q)-YPet foci reduced inappreciably from $2.6 \pm 0.4$ to $2.0 \pm 0.9$ foci per cell (Table 1), whereas frequency of transient foci remained unchanged from $2.4 \pm 0.5$ to $2.3 \pm 1$ foci per cell. Significantly, Mfd(E730Q) exhibited almost sixfold greater number of Rif-insensitive foci (compare transient binding events in Table 1: 2.4 foci per cell for Mfd(E730Q) vs. 0.4 foci per cell for wild-type Mfd). These experiments provide three findings: 1. Rif-insensitive foci can arise out of non-specific dsDNA binding; 2. TECs are not favoured substrates for binding of Mfd (E730Q) in vivo and 3. The ATP bound open configuration is not the predominant state of wild-type Mfd in Rif-treated cells.

We then set out to characterize the extent of RNAP-independent dsDNA binding of Mfd arising from stochastic transitions to the open configuration in its dynamic equilibrium. To that end, we probed the DNA binding ability of Mfd(L499R)-YPet which does not bind RNAP, and consequently does not exhibit long-lived foci (see Fig. 2). However, this mutant still exhibited transient foci that could arise either out of weak interactions with stalled TECs or from dsDNA binding. To assess whether these transient associations arose out of weak interactions with TECs, we imaged Mfd(L499R)-YPet in the presence of Rif. Mfd(L499R)-YPet exhibited only $0.2 \pm 0.1$ sub-second foci per cell in untreated cells (Table 1 and Fig. 5c). Moreover, Rif-treatment did not influence frequency of sub-second Mfd (L499R)-YPet binding, with $0.19 \pm 0.04$ foci per cell observed in Rif-treated cells (Table 1). The frequency of transient foci per cell was lower ($0.09 \pm 0.05$ foci per cell, see Table 1) and similarly insensitive to Rif-treatment (Fig. 5c), suggesting that Mfd(L499R) interactions possess a sub-second lifetime. These experiments allow us to rule out weak residual interactions with TECs as the causes of focus formation by this mutant. Invoking the known

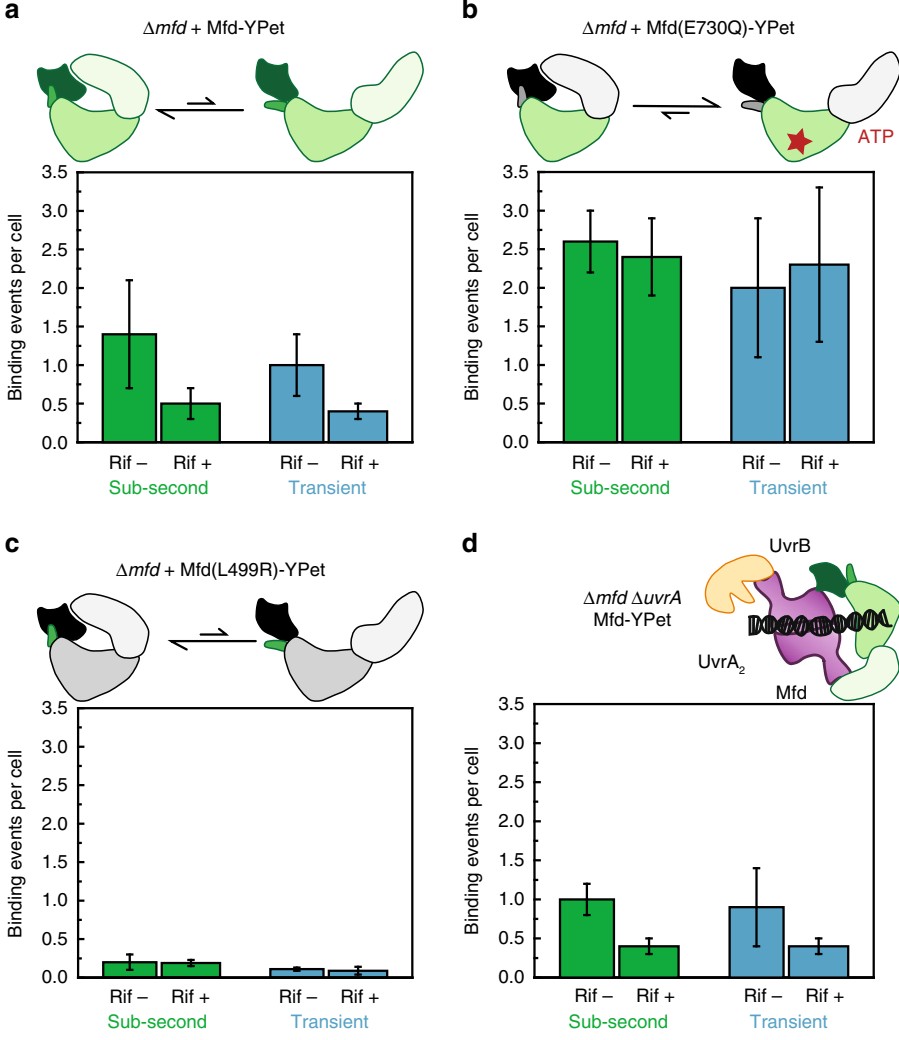

**Fig. 5** Influence of rifampicin on binding of plasmid-based Mfd-YPet and mutants. Bar plots represent sub-second (green) and transient (blue) binding events detected in **a**−**c** Δ*mfd* cells expressing **a** Mfd-YPet, **b** Mfd(E730Q)-YPet and **c** Mfd(L499R)-YPet, and in **d** Δ*mfd* Δ*uvr*A cells expressing Mfd-YPet. Cells grew in the absence (Rif−) or presence of Rif (Rif+, 50 µL per mL) for 1 h in the flow cell prior to imaging. Error bars are standard deviations from three experiments. **a**−**c** Cartoons illustrate the dynamic equilibrium of conformational states of Mfd and mutants. **d** The cartoon illustrates the hypothetical complex UvrB-UvrA$_2$-Mfd

activities of Mfd, the simplest explanation for the Mfd(L499R)-YPet foci is non-specific DNA binding.

However, this explanation alone cannot explain Mfd-YPet foci observed during Rif-treatment. When expressed to similar levels, Mfd-YPet forms between two- and fourfold more Rif-insensitive foci than the Mfd(L499R)-YPet foci on either timescale (Table 1 and Fig. 5c), leaving at least 0.3 foci per cell unaccounted for in cells expressing Mfd-YPet from the plasmid. We next investigated whether Mfd-UvrA intermediates represent the Rif-insensitive foci visualized in these experiments. Early experiments have revealed that high concentrations of UvrA can inhibit TCR[45], presumably by forming off-pathway intermediates that cannot engage TECs efficiently. Cytosolic Mfd exists in a dynamic equilibrium between the auto-inhibited state and the open configuration that can engage UvrA in the absence of stalled TECs[43]. At higher Mfd concentrations, futile Mfd-UvrA$_2$-Mfd/ UvrB intermediates could be formed, that would promote UvrA-mediated Rif-independent DNA binding. We examined this possibility by imaging plasmid-based Mfd-YPet in Δ*mfd* Δ*uvr*A cells. Deletion of *uvr*A led to a reduction in the frequency of Mfd-YPet sub-second foci per cell in untreated cells (compare 1.4 ± 0.7 to 1.0 ± 0.2 foci per cell, see Table 1 and Fig. 5a, d). However, Rif-

treated cells exhibited a comparable frequency of Rif-sensitive foci (compare 0.4 ± 0.1 *uvr*A$^−$ to 0.5 ± 0. 2 *uvr*A$^+$, see Table 1 and Fig. 5a, d). A model invoking an RNAP-independent, UvrA-mediated association of Mfd-YPet with the nucleoid fails to sufficiently explain the origins of the Rif-insensitive foci under these conditions.

## Discussion

In this work, we set out to visualize the first step of TCR. We employed single-molecule imaging approaches using a fluorescently labelled C-terminal fusion of Mfd, to visualize and quantify binding kinetics of Mfd in live cells. In the absence of external DNA damaging agents, we anticipated infrequent or transient associations of Mfd with RNAP in live cells, but were surprised to find that Mfd exhibits a stable interaction that lasts for several tens of seconds. Further investigation revealed that these stable binding events arise out of bona fide interactions with RNAP and UvrA. Four lines of evidence support this conclusion: first, mutation of the RNAP-interacting surface as in Mfd(L499R) results in the loss of long-lived foci. Second, 85% of Mfd foci in *mfd-ypet* cells are lost when transcription elongation is abolished

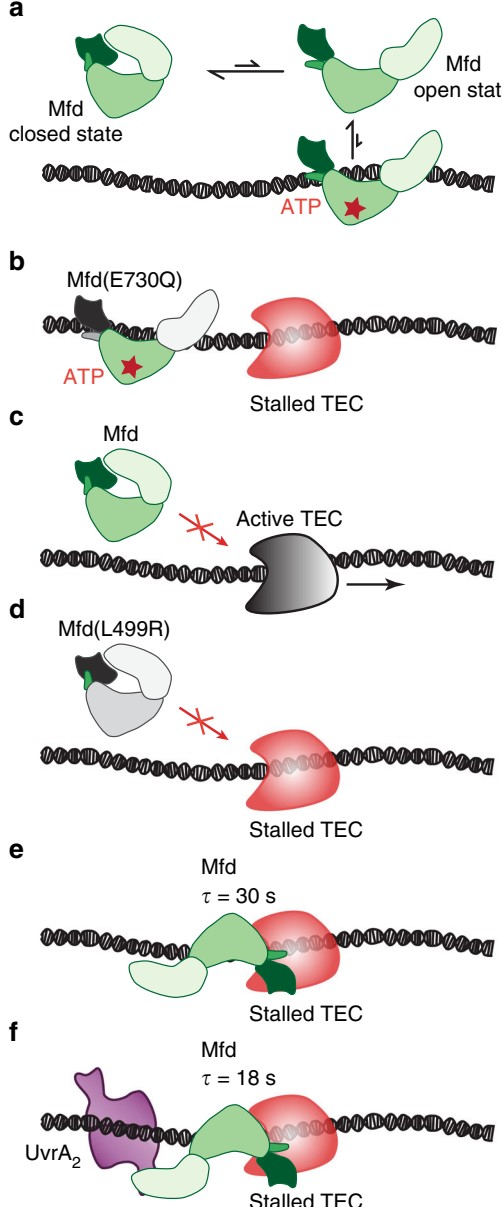

**Fig. 6** Multi-layered regulation of Mfd activities in vivo. **a** In solution, Mfd exists in a dynamic equilibrium between the closed state and the ATP-bound open state that is capable of short-lived DNA binding. **b** An Mfd mutant (E730Q) that is deficient in ATP hydrolysis exhibits non-specific DNA binding. **c** Active TECs are not a target for Mfd activity. Black arrow indicates the direction of TEC movement. **d** Stable association of Mfd and TEC is triggered by stalled TECs and involves the RNAP-interacting domain of Mfd. An Mfd mutant in which the RNAP-interacting domain is disrupted (L499R) loses the ability to engage stalled TECs. **e** Binding of Mfd is stimulated by stalled TECs, resulting in the formation of a long-lived complex with a lifetime of 30 s in the absence of UvrA. **f** UvrA$_2$ promotes the dissociation of Mfd from DNA, such that repair intermediates exhibit a lifetime of 18 s

in the presence of Rif. Third, treatment with CBR703—a drug that stalls RNAP—results in an increased number of stable binding events of Mfd. Finally, the lifetime of the stable interactions of Mfd is regulated by UvrA.

To probe the nature of the interaction of Mfd with RNAP, we treated cells with two small-molecule modulators of transcription —CBR703 and Rif. CBR703-treated cells exhibited a threefold

greater number of stable Mfd foci demonstrating that CBR703-stalled TECs are a target for Mfd. Rif-treatment of *mfd-ypet* cells (22 copies of Mfd per cell) led to an 85% loss of foci. Cells expressing Mfd-Ypet from a low copy plasmid (190 copies of Mfd-YPet per cell) exhibited a 64% loss of foci upon Rif-treatment. Stalled TECs thus represent the major substrate for Mfd activity.

Next, we harnessed the sensitivity of our approach to measure binding kinetics of Mfd in vivo. We employed an interval imaging strategy that enabled us to accurately measure binding kinetics of Mfd-YPet in growing cells. Mfd foci exhibited a mean lifetime of 18 s in the $uvr^+$ background. In contrast, $\Delta uvrA$ cells exhibited Mfd foci that dissociated with a mean lifetime of 30 s. By virtue of being RNAP- as well as, UvrA-dependent, this repair intermediate with an 18 s lifetime represents the lesion handover complex—encompassing the various stages of TCR from substrate recognition, RNAP displacement and lesion hand-over to either UvrA$_2$ or UvrA$_2$B. These measurements correspond well with previous measurements from in vitro single molecule experiments that measure the lifetimes of the Mfd-RNAP and Mfd-RNAP-UvrA(B) intermediates[17,42]. The identity of Mfd intermediates that dissociate with a 30 s lifetime in $\Delta uvrA$ cells remains less certain. These could potentially represent abortive intermediates or translocating Mfd following RNAP displacement where subsequent dissociation may occur stochastically[46] or upon encountering roadblocks[42].

In addition to detecting binding of Mfd to stalled TECs, we detected a small but significant fraction of binding events that were Rif-insensitive and dependent on Mfd concentration. To understand the origin of these binding events, we investigated two possibilities: first, Mfd binds dsDNA non-specifically, and second, Rif-insensitive binding events correspond to futile Mfd-UvrA$_2$-Mfd/UvrB intermediates. In a $\Delta uvrA$ background, the frequency of Rif-insensitive Mfd binding events was found to be unchanged compared to $uvrA^+$. This finding allowed us to rule out nonproductive UvrA-mediated complexes as the source of the short-lived interactions.

The observations of transient foci exhibited by the mutant Mfd (L499R) suggest that Mfd can bind dsDNA independently of RNAP in vivo. Mfd has been demonstrated to bind dsDNA in vitro[44]. Assuming the extents of non-specific dsDNA binding exhibited by wild-type Mfd and Mfd(L499R) are similar, an upper limit for non-specific dsDNA binding displayed by Mfd can be obtained from our experiments with Rif-treated cells. When normalized for the copy number of Mfd (190 copies per cell) vs. Mfd(L499R) (170 copies), 16% of sub-second foci could be attributed to non-specific dsDNA binding. However, this explanation still leaves 20% of the foci remain unaccounted for upon Rif treatment.

Rif-treatment does not result in a complete inhibition of RNAP association with the promoter[28,29]. A plausible source of these Rif-insensitive foci is Mfd interacting with promoter-associated transcription initiation complexes during Rif-treatment. Pull-down assays have showed that Mfd can bind to both the core RNAP and the holoenzyme (RNAP and σ subunit) with similar affinities in the absence of DNA[47,48]. Structural alignment of the RNAP initiation complex (PDB ID: 4YLN) with available structures of *Thermus thermophilus* TRCF (PDB ID: 3MLQ) and *E. coli* Mfd (PDB ID: 2EYQ) suggests that σ$^{70}$ and Mfd bind the upstream face of RNAP[14,49] in proximal, but non-overlapping sites (Supplementary Fig. 8). Specifically, the residues of β subunit (I117 K118 E119[48,50]) that are important for Mfd binding remain exposed and may permit simultaneous or competitive occupation of RNAP by σ$^{70}$ and Mfd[15]. While the promoter-proximal elongation complexes containing σ$^{70}$ were shown to be resistant to Mfd displacement[15], these results do not rule out transient

interactions between Mfd and RNAP-$\sigma^{70}$ complex on DNA. The extent to which $\sigma^{70}$ and Mfd simultaneously bind initiation complexes in vivo at promoter sites remains to be investigated.

Several elegant biochemical and structural investigations have suggested a 'spring-loaded' mechanism of action for Mfd function[14,16]. Our in vivo investigations support this model in which Mfd exists in a dynamic equilibrium between the closed, auto-inhibited conformation and the open conformation that is capable of DNA binding (Fig. 6a). The non-specific DNA binding of Mfd(E730Q) in Rif-treated cells supports the case for such a dynamic equilibrium, where ATP-bound Mfd(E730Q) is stabilized in the open conformation (Fig. 6b). The trigger for this spring-loaded mechanism is the binding of Mfd to stalled TECs (Fig. 6c) via the RNAP-interacting domain (Fig. 6d), resulting in the formation of the TEC bound open conformation in wild-type Mfd (Fig. 6e). Finally, a repair complex containing UvrA promotes dissociation of Mfd from DNA (Fig. 6f).

## Methods

**Strains and plasmids**. All bacterial strains used were derivatives of E. coli K-12 MG1655. Strains were constructed using λ-Red recombination, P1 transduction or obtained as listed in Supplementary Table 1. Plasmids carrying *mfd* promoter and allele for Mfd-YPet or mutants are listed in Supplementary Table 2. Oligonucleotides used in λ-Red recombination and cloning are listed in Supplementary Table 3.

**UV survival assay**. ΔrecA (HH021), Δmfd ΔrecA (HH023) and mfd-ypet ΔrecA (HH034) were streaked on Luria-Bertani (LB) plates and grown overnight at 37 °C from −80 °C stocks. For each strain, cells from a single colony were incubated in 1 mL of LB liquid media in 2 mL microcentrifuge tubes, at 37 °C for 24 h and shaken at 1000 rpm (Eppendorf ThermoMixer C, Eppendorf, Germany). 10 μL of the stationary phase culture were used to inoculate 1 mL of LB liquid media, followed by incubation for 2 h. Early exponential phase (OD$_{600}$ 0.2–0.3) cultures were then centrifuged at $3000 \times g$ (Eppendorf Microcentrifuge 5424, Eppendorf, Germany) for 5 min at room temperature. The cell pellets were washed twice with 1 mL of 0.1 M MgSO$_4$ solution. The pellets were resuspended in 0.1 M MgSO$_4$ to obtain an OD$_{600}$ of 0.25.

20 μL (~ $4 \times 10^6$ cells) of cells were transferred to 96-well plates and irradiated (Herolab UV-8 SL, Herolab, Germany) with 254 nm-UV light at doses of 0, 1, 2.5 and 5 J m$^{-2}$ respectively. 5 μL (containing ~$10^6$ cells, $N_0$) of the UV-irradiated cell suspension were used to inoculate 15 mL of LB liquid media in 50 mL Falcon tubes (in duplicate and incubated at 37 °C with shaking at 200 rpm. OD$_{600}$ was monitored for every hour from 7 to 24 h. At each time point, the average was taken from two technical replicates and converted to the total amounts of cells ($N$), assuming OD$_{600}$ of 1.0 corresponds to $8 \times 10^8$ cells per mL. From two independent experiments, the mean ln($N/N_0$) and standard error of the mean were plotted in Fig. 1b.

**Flow-cell assembly**. Live-cell imaging was performed in home-built flow cells as described previously[51]. Briefly, the flow cell was assembled from a quartz top piece (45 × 20 × 1 mm, ProSciTech, Australia) and an APTES-treated glass coverslip (24 × 50 mm, Australian Scientific) using double-sided sticky tape (3M). Coverslips were functionalized by sonicating for 30 min with 5 M KOH, followed by extensive rinsing with Milli-Q water and incubating in solution of 5% (v/v) of APTES ((3-aminopropyl)-triethoxysilane 98%, Alfa Aesar, USA) in Milli-Q water for 5 min. The coverslips were then washed once with ethanol and sonicated in ethanol for 1 min. Finally, functionalized coverslips were washed with Milli-Q water and rapidly dried with compressed nitrogen.

**Cell culture for imaging**. Cells were grown at 30 °C in EZ-rich defined media (Teknova, CA, USA) supplemented with 0.2% (w/v) glucose. For experiments involved plasmid-based expression of Mfd-YPet and mutants, spectinomycin (50 μg per mL, Sigma-Aldrich, USA) was added to the growth media. Cultures at early exponential phase were loaded into a custom-built flow cell maintained at 30 °C. Aerated growth media was supplied during the experiment at a rate of 30 μL per min using a syringe pump (Adelab Scientific, Australia). For experiments involving drug treatment, growth medium was supplemented with rifampicin (50 μg per mL, Sigma-Aldrich, USA) or CBR703 (75 μg per mL, Thermo Fisher Scientific, UK). The minimal inhibitory concentrations of rifampicin for E. coli K-12 and CBR703 for E. coli tolC$^-$ were determined to be 7 μg per mL and 14 μg per mL respectively[52].

**Fluorescence microscope and imaging**. Single-molecule fluorescence imaging was performed with a custom-built microscope (Supplementary Fig. 1), operating in HILO mode[53]. The microscope was constructed with an inverted microscope body (Nikon Eclipse-Ti, Nikon, Japan) equipped with a 1.49 NA 100× objective and a 512 × 512 pixel$^2$ Photometrics Evolve CCD camera (Photometrics, Arizona, USA). NIS-Elements equipped with JOBS module (Nikon, Japan) was used to operate the microscope. Mfd-YPet and mutants were imaged with a 514-nm Sapphire LP laser (150 mW max. output, Coherent, CA, USA), and ET535/30m emission filter (Chroma, Vermont, USA). The operating laser power density measured directly at the sample above the objective lens with the laser pointing up was 71 W cm$^{-2}$. RpoC-PAmCherry was imaged by first activating using a 405-nm OBIS laser (200 mW max. output, Coherent, CA, USA) followed by read-out using a 568-nm Sapphire LP laser (200 mW max. output, Coherent, CA, USA). PAm-Cherry emission was collected using a ET590lp filter (Chroma, Vermont, USA). The operating power densities for 405-nm and 568-nm lasers were 25 W cm$^{-2}$ and 442 W cm$^{-2}$ respectively.

Continuous imaging was acquired at a rate of 10 fps for 5 or 15 s. Interval imaging of Mfd-YPet (*mfd-ypet uvrA$^+$* and *mfd-ypet ΔuvrA*) was performed in two phases. In the first phase, 50 frames were collected as rapid acquisitions at a frame rate of $1/\tau_{int}$ ($\tau_{int} = 0.1$ s). In the second phase, 100 frames were collected at a frame rate ($1/\tau_{tl}$). Where $\tau_{tl}$ is the sum of the integration time ($\tau_{int}$; 0.1 s) and a fixed dark time ($\tau_d$; $\tau_{tl} = \tau_{int} + \tau_d$). In each experiment, videos with varying $\tau_{tl}$ (ranging from 0.1 to 10 s, see Fig. 4b and Supplementary Tables 4,5) were acquired. To minimize laser damage, we ensured that a new set of cells was imaged for each different value of the parameter $\tau_{tl}$. For measurements of sub-second or transient foci per cell (Table 1 and Fig. 5), the two-phase imaging protocol was employed. The first phase included 100-frame rapid acquisition (10 fps) while the second phase contained 100 frames (0.1 s each) acquired with $\tau_{tl}$ of 0.1 or 1 s.

RpoC-PAmCherry was activated with the 405-nm laser for 1 s, and rapid acquisitions (10 fps) were acquired in 568-nm channel for 5 s (Supplementary Fig. 6).

**Image analysis**. Image analysis was performed in Fiji[54], using the Single Molecule Biophysics plugins (available at https://github.com/SingleMolecule/smb-plugins). Raw videos (nd2 format) were converted into TIF files using Bio-Formats plugin, flattened with excitation beam profile as described previously[51]. Cell outlines were drawn in MicrobeTracker[55]. For quantification of binding events, peaks were detected in the reactivation phase (Fig. 3f: frames 31−50, Fig. 4b: frames 51−100, Table 1 and Fig. 5: frames 101−200). Peaks were detected by first applying a discoidal average filter (inner radius of one pixel, outer radius of three pixels), then selecting pixels above the intensity threshold (mean + 10× s.d.). Secondly, only peaks with the full width at half maximum smaller than five pixels (530 nm) were kept to reject broad cytosolic features. Peaks detected within two-pixel radius (213 nm) in consecutive frames were considered to belong to the same binding event.

**Dissociation kinetics of Mfd**. A cumulative residence time distribution of binding events as a function of frame-time was compiled for each $\tau_{tl}$ from nine and ten independent experiments for *mfd-ypet uvrA$^+$* and *mfd-ypet ΔuvrA* respectively (Supplementary Tables 4,5). Frame-time was converted to real time using the formula $t = (n − 1)\tau_{tl}$ with $n$ being the number of frames for which a focus could be detected at the same location (Fig. 4b). The effective off-rate constant $k_{eff}$, representing a mixture of $k_b$ and $k_{off}$, was obtained by fitting the cumulative residence time distribution to a single-exponential model as in Eq. (1):

$$f_1(t) = A \exp(-k_{eff}t) = A \exp\left(-\left(k_b \frac{\tau_{int}}{\tau_{tl}} + k_{off}\right)t\right), \quad (1)$$

where $A$ is the number of molecules, $t$ is real time in seconds[39]. To determine uncertainties in $k_{eff}\tau_{tl}$, fitting was performed a thousand times on cumulative residence time distributions derived from randomly selecting 80% of the binding events (bootstrapping and single-exponential fitting using custom-written MATLAB codes, MathWorks, USA).

Equation (1) can be rearranged to yield a linear relationship between $k_{eff}\tau_{tl}$ and $\tau_{tl}$ (Eq. (2)):

$$k_{eff}\tau_{tl} = k_b\tau_{int} + k_{off}\tau_{tl}. \quad (2)$$

**Data availability**. The data that support the findings of this study and the codes used for data analysis are available from the corresponding author upon request.

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

## Acknowledgements

We thank Nigel Savery for providing us with the gene for Mfd(L499R), Mike Heilemann for providing us with the strain MG1655 *rpoC-PAmCherry*, CGSC for *E. coli* Δ*tolC* strain, Elizabeth Wood and Michael Cox for P1 lysate, Anuk Indraratna and Kasey Markel for assistance with cloning and P1 transduction respectively. We thank Andrew Robinson for useful comments. A.M. van Oijen would like to acknowledge support by the Australian Research Council (DP150100956 and FL140100027).

## Author contributions

Construct creation: H.N.H.; Data curation: H.N.H.; Data analysis: H.N.H. and H.G.; Software: H.N.H. and H.G.; Writing—Original draft: H.N.H. and H.G.; Writing—review and editing: H.G. and A.M.v.O.; Conceptualization: H.G.; Supervision: H.G. and A.M.v.O.

## Additional information

**Competing interests:** The authors declare no competing interests.

