## [Peer Review File · Nature Communications]

Reviewers' comments:

Reviewer #1 (Remarks to the Author):

This elegant manuscript by Ho et al. uses in vivo imaging of individual fluorescent Mfd molecules to learn about the role and dynamics of Mfd in vivo. Imaging of wildtype Mfd shows it to be expressed at only ~22 copies/cell (lower than previously determined) and to localize to foci, and comparison with a mutant of Mfd (L499R) unable to interact with RNA polymerase indicates that wildtype Mfd foci correspond to the protein associating with RNAP. Rif-based inhibition of transcription abolishes foci, whereas CBR703-based induction of RNA polymerase stalling increased foci. No foci were observed when RNA polymerase stalling was induced by CBR703 but Mfd-L499R was used. It appears the foci are likely due to RNA polymerase clustering on highly active genes, and therefore the recruitment of multiple Mfds to these areas where RNAP density is elevated. Finally, the authors determine that the residence time of Mfd in these foci is about 18 seconds.

Although it is important to observe Mfd interacting with RNA polymerase in vivo, it cannot be said that much of the results presented are novel. The genetics, structural biology, and biochemistry have firmly established over the course of many decades that Mfd indeed interacts with RNAP to displace it from DNA lesions and then recruit Uvr proteins. Prior work has also firmly and fully established that Mfd is responsible for the rapid recovery of transcription in UV-exposed cells before the SOS-response. Therefore the novel finding in this work is the determination in vivo of the duration of the Mfd interactions with RNA polymerase at ~18 s. Unfortunately this single number alone is of limited use as it cannot be compared to anything to determine either consistency or novelty.

There is no guarantee that this time corresponds to successful interactions between the two proteins leading to dissociation of RNA polymerase. It is in reality likely to be a combination of 1) binding events which do not lead to displacement of RNA polymerase but instead spontaneously dissociate and 2) binding events which do lead to displacement of RNA polymerase. To make a contribution to the field that would warrant publication in Nature Communications, it is this reviewer's position that more work is required. To prove that this 18-second time actually includes RNAP displacement, the authors could carry out an experiment using an Mfd mutant deficient in ATP hydrolysis. To prove that this time

actually includes successful lesion handoff to the Uvr system, the authors could carry out an experiment using an Mfd mutant deficient in the UvrB homology module. With the current results as a base these additional measurements would represent an important advance.

Reviewer #2 (Remarks to the Author):

Ho et al. report observations and characterization of the behavior of the transcription-coupled repair factor Mfd in live *E. coli* cells using Mfd tagged with a bright fluorescent protein. By applying state-of-the-art single-molecule microscopy, a mutant Mfd unable to bind RNA polymerase, and small-molecule inhibitors of transcription initiation or elongation, the authors establish that Mfd binds elongation complexes (ECs) in cells, measure the *in vivo* off-rate for Mfd-EC interaction, and show that Mfd interacts with ECs even under non-DNA-damage conditions. The experiments are expertly performed and the manuscript is written in a readily accessible style. The results will be of interest to a broad spectrum of researchers from bacterial physiology to transcriptional and DNA repair mechanisms to single-molecule biophysics. I enjoyed reading the manuscript and congratulate the authors on the excellent work. I have only minor suggestions for the authors to consider in preparing a final version of the manuscript.

1. The observation that Mfd does not associate with the nucleoid in Rif-treated cells or in *mfd-L499R* mutant cells is interpreted to mean that Mfd associates with only through its interaction with RNA polymerase. That is a reasonable model, but an alternative would be that Mfd scans DNA rapidly through actions of its DNA-binding and motor domains so that no foci are evident. One way to distinguish such a model would be to show that there is no concentration of Mfd in nucleoid vs. non-nucleoid regions of the cell by careful quantitation of the distribution of signal in Rif-treated or *mfd-L499R* mutant cells. Is there even weak concentration in the nucleoid? The images presented by the authors suggest the answer is no, but a more careful quantitation of the signal distribution across the cellular volume would be more convincing and is worth including in this publication.

2. Related to this suggestion, however, why is Mfd association with the nucleoid lost upon Rif treatment? Mfd is reported to bind sigma70-containing RNA polymerase as tightly as core RNA polymerase (Smith and Savery, *NAR* 33:755, 2005) . Given that Rif locks sigma70-RNA polymerase onto promoter DNA, what prevents Mfd from associating with the promoter-bound RNA polymerase? It would be helpful to readers to address the question explicitly; for instance if there are published experiments that explain the loss of Mfd association, these could be cited. Alternatively, if this is a novel discovery, then it deserves more comment.

3. On page 3, second paragraph, the authors state that Mfd is thought to be present at ~500 copies per cell, but that they measure the Mfd-YPet level to be ~22 copies per cell. What is the

explanation for this ~20x discrepancy? This question should be addressed explicitly in a revised manuscript.

4. On page 3, first paragraph, the authors state that the survival of *mfd*-ypec Δ recA cells upon UV treatment was comparable to that of *mfd*+ Δ recA cells. However, in Fig 1b panel 4, the *mfd*-tagged cells appear to require a longer time to recover from UV treatment. Why is this consistent with an interpretation that survival was comparable (presumably meaning similar)? Is it possible that the lower level of tagged Mfd protein (see point 3) might explain the compromised recovery? This discrepancy should be addressed explicitly in a revised manuscript.

Response to the reviewers

We thank the reviewers for their constructive feedback. To address the concerns raised by the reviewers, we have performed additional experiments and analyses, and found these tremendously helpful to strengthen our study and to provide further mechanistic insight.

Reviewer #1 (Remarks to the Author):

This elegant manuscript by Ho et al. uses in vivo imaging of individual fluorescent Mfd molecules to learn about the role and dynamics of Mfd in vivo. Imaging of wildtype Mfd shows it to be expressed at only ~22 copies/cell (lower than previously determined) and to localize to foci, and comparison with a mutant of Mfd (L499R) unable to interact with RNA polymerase indicates that wildtype Mfd foci correspond to the protein associating with RNAP. Rif-based inhibition of transcription abolishes foci, whereas CBR703-based induction of RNA polymerase stalling increased foci. No foci were observed when RNA polymerase stalling was induced by CBR703 but Mfd-L499R was used. It appears the foci are likely due to RNA polymerase clustering on highly active genes, and therefore the recruitment of multiple Mfds to these areas where RNAP density is elevated. Finally, the authors determine that the residence time of Mfd in these foci is about 18 seconds.

Although it is important to observe Mfd interacting with RNA polymerase in vivo, it cannot be said that much of the results presented are novel. The genetics, structural biology, and biochemistry have firmly established over the course of many decades that Mfd indeed interacts with RNAP to displace it from DNA lesions and then recruit Uvr proteins. Prior work has also firmly and fully established that Mfd is responsible for the rapid recovery of transcription in UV-exposed cells before the SOS-response. Therefore the novel finding in this work is the determination in vivo of the duration of the Mfd interactions with RNA polymerase at ~18 s. Unfortunately this single number alone is of limited use as it cannot be compared to anything to determine either consistency or novelty.

There is no guarantee that this time corresponds to successful interactions between the two proteins leading to dissociation of RNA polymerase. It is in reality likely to be a combination of 1) binding events which do not lead to displacement of RNA polymerase but instead spontaneously dissociate and 2) binding events which do lead to displacement of RNA polymerase. To make a contribution to the field that would warrant publication in Nature Communications, it is this reviewer's position that more work is required. To prove that this 18-second time actually includes RNAP displacement, the authors could carry out an experiment using an Mfd mutant deficient in ATP hydrolysis. To prove that this time actually includes successful lesion handoff to the Uvr system, the authors could carry out an experiment using an Mfd mutant deficient in the UvrB homology module. With the current results as a base these additional measurements would represent an important advance.

We thank the reviewer for their positive comments and suggestions for two experiments that would provide more mechanistic context to the duration of the Mfd-RNA Polymerase interaction observed by us *in vivo*. We pursued both directions and are pleased to add the results to our manuscript, making it significantly stronger. We agree with the reviewer that the 18 s lifetime, by itself, can

either represent unproductive engagement with RNAP, or productive engagement with RNAP that results in RNAP displacement and potentially followed by UvrA or UvrAB recruitment.

Per this reviewer's request to shed light on the influence of the ATPase activity of Mfd on its residence time *in vivo*, we constructed a low-copy plasmid expressing Mfd(E730Q)-YPet and visualized it in Δmfd cells (Supplementary Fig. 3d). Mfd(E730Q) is a mutant that is deficient in ATP hydrolysis but not ATP binding (Deaconescu et al. 2012). We found the expression level of Mfd(E730Q)-YPet in these cells to be about 140 copies/cell (now included as Supplementary Fig. 2f). Surprisingly, single-molecule live-cell imaging of Mfd(E730Q)-YPet revealed that: 1. This mutant forms foci twice as frequently as Mfd-YPet 2. Focus formation is independent of Rif-treatment. A previous study has demonstrated that Mfd and Mfd(E730Q) bind dsDNA non-specifically in the presence of the non-hydrolysable nucleotide ATP γ S. Since Mfd(E730Q) is deficient in ATP hydrolysis, the Rif-insensitive foci are likely dsDNA binding events. Experiments with Rif reveal that transcription elongation complexes are not the primary substrate of Mfd(E730Q)-YPet. This result is now included in included in section titled 'Mfd associates with sites other than stalled TECs *in vivo*' and Figure 5.

This reviewer also suggested using a mutant Mfd that does not engage UvrA to unveil the significance of the measured lifetime. Since the introduction of mutations in Mfd can deregulate its catalytic functions (Manelyte et al. 2010, as also in the case of E730Q), we instead chose to examine Mfd-YPet kinetics in $\Delta uvrA$ cells to gain an understanding of the relevance of the measured 18 s lifetime. This background enables us to interrogate Mfd binding kinetics in cells where stalled TECs may be recognized by Mfd but lesion-handover to the NER pathway is impossible. Remarkably, Mfd-YPet was found to dissociate slower in this background, with measured lifetime around 30 s. The finding that UvrA promotes the dissociation of Mfd on DNA is consistent with single-molecule TCR reconstitution experiments from the Strick lab. These experiments prove that the measured lifetime of Mfd-YPet is influenced both by RNAP as well as, UvrA. We therefore interpret the 18 s lifetime to encompass RNAP engagement, displacement and coupling to the NER factors UvrA(B). This result yields important mechanistic insights into the recruitment of transcription-coupled repair machineries in live cells. Further, an important finding from our work is that the lifetime of Mfd foci *in vivo* is approximately 5X faster than the corresponding *in vitro* experiments described by Fan et al., Nature, 2016. This result is now included in section 'Interactions of Mfd with TECs are long-lived'.

Reviewer #2 (Remarks to the Author):

Ho et al. report observations and characterization of the behavior of the transcription-coupled repair factor Mfd in live E. coli cells using Mfd tagged with a bright fluorescent protein. By applying state-of-the-art single-molecule microscopy, a mutant Mfd unable to bind RNA polymerase, and small-molecule inhibitors of transcription initiation or elongation, the authors establish that Mfd binds elongation complexes (ECs) in cells, measure the in vivo off-rate for Mfd-EC interaction, and show that Mfd interacts with ECs even under non-DNA-damage conditions. The experiments are expertly performed and the manuscript is written in a readily accessible style. The results will be of interest to a broad spectrum of researchers from bacterial physiology to transcriptional and DNA repair mechanisms to single-molecule biophysics. I enjoyed reading the manuscript and congratulate the authors on the excellent work. I have only minor suggestions for the authors to consider in preparing a final version of the manuscript.

1. The observation that Mfd does not associate with the nucleoid in Rif-treated cells or in mfd-L499R mutant cells is interpreted to mean that Mfd associates with only through its interaction with RNA polymerase. That is a reasonable model, but an alternative would be that Mfd scans DNA rapidly through actions of its DNA-binding and motor domains so that no foci are evident. One way to distinguish such a model would be to show that there is no concentration of Mfd in nucleoid vs. non-nucleoid regions of the cell by careful quantitation of the distribution of signal in Rif-treated or mfd-L499R mutant cells. Is there even weak concentration in the nucleoid? The images presented by the authors suggest the answer is no, but a more careful quantitation of the signal distribution across the cellular volume would be more convincing and is worth including in this publication.

We agree with the reviewer that a plausible explanation for the low frequency of foci in Rif-treated cells or exhibited by Mfd(L499R)-YPet could be these molecules rapidly scan DNA. With our imaging condition using 100 ms exposures, we are unable to distinguish if these molecules are freely diffusive or weakly associate with the nucleoid while scanning. However, further characterization of Mfd(L499R)-YPet and Mfd-YPet in Rif-treated cells supports the case for a low level of association with the nucleoid:

1. By single-molecule tracking and quantifying binding events lasting for at least 0.2 s, we observed about 0.2 binding events per cell of Mfd(L499R)-YPet in both untreated or Rif-treated cells. In comparison, control studies with plasmid-based Mfd-YPet yield 1.4 and 0.5 binding events per cell in untreated and Rif-treated cells respectively. As the point mutation L499R had been firmly established to disrupt the Mfd-RNAP interaction, we interpret the low frequency of binding events exhibited by Mfd(L499R) may represent dsDNA binding independent of RNAP.
2. Rif-treatment causes Mfd-YPet signal to be more homogenous, but a small fraction of foci (~14%) persists during Rif-treatment. These may represent binding to dsDNA or interaction with transcription initiation complexes. At nine-fold higher Mfd-YPet concentration (plasmid-based Mfd-YPet), the fraction of Rif-insensitive foci increases to 36%. This suggests Mfd binds a secondary substrate and this binding is dependent on the concentration of Mfd (see also response to comment #2 below).

These results are included in section 'Mfd associates with sites other than stalled TECs in vivo' and Figure 5.

2. Related to this suggestion, however, why is Mfd association with the nucleoid lost upon Rif treatment? Mfd is reported to bind sigma70-containing RNA polymerase as tightly as core RNA polymerase (Smith and Savery, NAR 33:755, 2005) . Given that Rif locks sigma70-RNA polymerase onto promoter DNA, what prevents Mfd from associating with the promoter-bound RNA polymerase? It would be helpful to readers to address the question explicitly; for instance if there are published experiments that explain the loss of Mfd association, these could be cited. Alternatively, if this is a novel discovery, then it deserves more comment.

We attributed the loss of stable Mfd association with the nucleoid in the presence of Rif to loss of TCR complexes at sites of stalled TECs. As the reviewer points out, we cannot rule out that the short-lived associations observed in the presence of Rif could arise from binding of Mfd to initiation complexes. Consistent with this hypothesis, we found Mfd-YPet exhibit a small fraction of Rif-insensitive foci and the percentage of Rif-insensitive foci increases 2.3 fold when Mfd-YPet expression level rises nine-fold. To understand this binding better, we investigated two additional scenarios: 1. that Mfd binds dsDNA non-specifically and 2. Futile repair intermediates including UvrA could retain DNA binding and consequently manifest as foci.

1. We obtained an upper limit on the fraction of dsDNA binding complexes of Mfd, by examining binding of Mfd(L499R) in cells. Our studies revealed that only 44% (Table 1 maintext) of Rif-insensitive foci could be explained by dsDNA binding that is independent of RNAP.
2. Mfd focus formation in *uvrA*⁺ cells did not change appreciably in *uvrA*⁻ cells.

In light of these findings, the hypothesis that Mfd and σ^{70} can simultaneously engage promoter bound RNAP becomes increasingly plausible. As the reviewer points out, previous work has demonstrated that σ^{70} and Mfd can both bind RNAP in solution. A structural alignment of Mfd with promoter bound holoenzyme (now included as Supplementary Fig. 8 and main text) reveals that the binding sites of Mfd and σ^{70} are proximal but non-overlapping and residues of β subunit interacting with Mfd are exposed. Based on this structural alignment, simultaneous binding of Mfd and σ^{70} to the core polymerase is plausible. In the absence of a lesion in promoter DNA, the lifetime of gratuitous Mfd association could be short-lived, and could very well explain the Rif-insensitive foci observed in our experiments. Whether Mfd can compete with σ^{70} and regulate transcription is beyond the scope of this work and represents an important question that awaits future investigation.

3. On page 3, second paragraph, the authors state that Mfd is thought to be present at ~500 copies per cell, but that they measure the Mfd-YPet level to be ~22 copies per cell. What is the explanation for this ~20x discrepancy? This question should be addressed explicitly in a revised manuscript.

The discrepancy in copy number between our measurement of Mfd-YPet (measured by fluorescence intensity of single cells) and in literature is now addressed in 'Results' (lines 100-107). The widely accepted value of 500 Mfd copies per cell was presumably derived from Western blot data that

remains unpublished (data not shown). In the absence of high-quality Mfd antibody, we are unable to reproduce this assay. On the other hand, our measurement of Mfd-YPet copy number is based on the fluorescence intensity from single cells, which may represent underestimation of true Mfd copy number due to the slow maturation or misfolding of the tag, or lower transcription and translation efficiencies due to the C-terminal tag.

*4. On page 3, first paragraph, the authors state that the survival of *mfd-ypec ΔrecA* cells upon UV treatment was comparable to that of *mfd+ ΔrecA* cells. However, in Fig 1b panel 4, the *mfd*-tagged cells appear to require a longer time to recover from UV treatment. Why is this consistent with an interpretation that survival was comparable (presumably meaning similar)? Is it possible that the lower level of tagged Mfd protein (see point 3) might explain the compromised recovery? This discrepancy should be addressed explicitly in a revised manuscript.*

It is possible that the slight delays in recovery of *mfd-ypec* cells compared to wildtype at 2.5 and 5 J/m² UV doses may be a result of lower Mfd-YPet copy number compared to wildtype Mfd. We have changed the statement that 'survival was comparable' to *mfd-ypec* cells exhibited a slight delay following exposures to UV doses of 2.5 and 5 J/m² (line 79-82).

Reviewers' Comments:

Reviewer #1 (Remarks to the Author):

The authors have revised their manuscript with important new experiments and data and have addressed my main concerns. The discussion of the solution-based conformational equilibrium of Mfd is interesting in light of recent work from the Wang lab. The revised manuscript warrants publication in Nature Communications.

The idea under discussion with referee 2 that Mfd interacts with sigma70-containing RNA polymerase is a bit surprising given the reported inability of Mfd to displace sigma70-containing RNAP from DNA (Park, Marr and Roberts, Cell 2002). If this discussion remains as it is (pg. 10 line 359) I recommend it at least be clarified that this is not in agreement with prior results from the Roberts lab.

Reviewer #2 (Remarks to the Author):

Ho et al. have revised their manuscript in response to reviewer critiques. In my opinion, they have done an admirable job of addressing the reviewers' concerns in this revision, including by addition of important new experimental findings. I believe the manuscript is now acceptable for publication.

I have one suggestion for the authors as they prepare a final version of the manuscript, assuming that it is accepted. First, on the issue of copy number of Mfd, there are some good measurements available from quantitative mass spec and available in the paper Schmidt et al., 2016. The quantitative and condition-dependent Escherichia coli proteome. Nat Biotechnol 34, 104-110. The Schmidt et al numbers place Mfd in the 60-100 copies/cell range, depending on growth medium and between the YPet measurements and the earlier Western blot-based measurements. Given the concern about underestimation due to slow maturation of YPet perhaps the authors data agree with the Schmidt et al. measurements. If so, it would be worth citing the Schmidt et al. paper to reinforce the point that 500 copies per cell is likely an overestimate. Additionally, however, this revision might also require a revision of the estimate of Mfd copy number generated by plasmid based overexpression. Perhaps that number should be expressed as a fold increase relative to nature Mfd abundance, to avoid any understatement due to slow YPet maturation.

REVIEWERS' COMMENTS:

Reviewer #1 (Remarks to the Author):

The authors have revised their manuscript with important new experiments and data and have addressed my main concerns. The discussion of the solution-based conformational equilibrium of Mfd is interesting in light of recent work from the Wang lab. The revised manuscript warrants publication in Nature Communications.

The idea under discussion with referee 2 that Mfd interacts with sigma70-containing RNA polymerase is a bit surprising given the reported inability of Mfd to displace sigma70-containing RNAP from DNA (Park, Marr and Roberts, Cell 2002). If this discussion remains as it is (pg. 10 line 359) I recommend it at least be clarified that this is not in agreement with prior results from the Roberts lab.

Per the reviewer's suggestion, we now included the finding from the Roberts lab (Park et al. 2002) in the discussion (lines 351-353, No Markup mode). In Park et al. 2002, the authors used a promoter-proximal elongation complex containing sigma70 and demonstrated Mfd inability to displace this RNAP-sigma70 complex. However, this result does not rule out unproductive transient interactions between Mfd and RNAP. Additionally, it is likely that initiation complex (including Rif-stalled initiation complex) adopts a different conformation than that of promoter-proximal elongation complex, possibly allowing competitive binding of sigma70 and Mfd (see Supplementary Fig 8).

Reviewer #2 (Remarks to the Author):

Ho et al. have revised their manuscript in response to reviewer critiques. In my opinion, they have done an admirable job of addressing the reviewers' concerns in this revision, including by addition of important new experimental findings. I believe the manuscript is now acceptable for publication.

I have one suggestion for the authors as they prepare a final version of the manuscript, assuming that it is accepted. First, on the issue of copy number of Mfd, there are some good measurements available from quantitative mass spec and available in the paper Schmidt et al., 2016. The quantitative and condition-dependent *Escherichia coli* proteome. Nat Biotechnol 34, 104-110. The Schmidt et al numbers place Mfd in the 60-100 copies/cell range, depending on growth medium and between the YPet measurements and the earlier Western blot-based measurements. Given the concern about underestimation due to slow maturation of YPet perhaps the authors data agree with the Schmidt et al. measurements. If so, it would be worth citing the Schmidt et al. paper to reinforce the point that 500 copies per cell is likely an overestimate. Additionally, however, this revision might also require a revision of the estimate of Mfd copy number generated by plasmid based overexpression. Perhaps that number should be expressed as a fold increase relative to nature Mfd abundance, to avoid any understatement due to slow YPet maturation.

We thank the reviewer for directing our attention to Schmidt et al. 2016. Mfd copy numbers were determined to be 48 and 34 per *E. coli* MG1655 cell grown in LB and minimal medium supplemented with glucose respectively. These measurements are largely consistent with our estimate of Mfd copy number (22 per cell). These values are now cited in the Result section (lines 91-94, No Markup mode).

Additional comments:

1. We have also added an additional discussion reconciling our measurements with the previous work by the Strick work (lines 322-324, No Markup mode).
2. We have changed the statement “While preparing this manuscript, a recent study demonstrated that Mfd is capable of translocating on naked dsDNA templates in the absence of RNAP⁴⁶” in the discussion to “Mfd has been demonstrated to bind dsDNA *in vitro*⁴⁴” (line 336, No Markup mode) to cite a seminal paper that is a more appropriate reference in the context of the DNA binding properties of Mfd.

These changes do not influence the results or interpretations of this work.